# Light-Emitting Biosilica by In Vivo Functionalization of *Phaeodactylum tricornutum* Diatom Microalgae with Organometallic Complexes

**Danilo Vona [1], Roberta Ragni [1,*], Emiliano Altamura [1] , Paola Albanese [1] , Maria Michela Giangregorio [2], Stefania Roberta Cicco [3] and Gianluca Maria Farinola [1,*]**

[1] Chemistry Department, University of Bari "Aldo Moro", Via Orabona 4, 70126 Bari, Italy; danilo.vona@uniba.it (D.V.); emiliano.altamura@uniba.it (E.A.); paola.albanese@uniba.it (P.A.)

[2] Chemistry Department, Consiglio Nazionale delle Ricerche-Istituto di Nanotecnologia (CNR-Nanotec), Via Orabona 4, 70126 Bari, Italy; michelaria.giangregorio@nanotec.cnr.it

[3] Chemistry Department, Consiglio Nazionale delle Ricerche-Istituto di Chimica dei Composti Organo Metallici (CNR-ICCOM), Via Orabona 4, 70126 Bari, Italy; cicco@ba.iccom.cnr.it

**\*** Correspondence: roberta.ragni@uniba.it (R.R.); gianlucamaria.farinola@uniba.it (G.M.F.)

**Abstract:** In vivo incorporation of a series of organometallic photoluminescent complexes in *Phaeodactylum tricornutum* diatom shells (frustules) is investigated as a biotechnological route to luminescent biosilica nanostructures. $[Ir(ppy)_2bpy]^+[PF_6]^-$, [(2,2′-bipyridine)bis(2-phenylpyridinato) iridium(III) hexafluorophosphate], $[Ru(bpy)_3]^{2+}$ $2[PF_6]^-$, [tris(2,2′-bipyridine)ruthenium(II) hexafluorophosphate], $AlQ_3$ (tris-(8-hydroxyquinoline)aluminum), and $ZnQ_2$ (bis-8-hydroxyquinoline-zinc) are used as model complexes to explore the potentiality and generality of the investigated process. The luminescent complexes are added to the diatom culture, and the resulting luminescent silica nanostructures are isolated by an acid-oxidative treatment that removes the organic cell matter without altering both frustule morphology and photoluminescence of incorporated emitters. Results show that, except for $ZnQ_2$, the protocol successfully leads to the incorporation of complexes into the biosilica. The spontaneous self-adhering ability of both bare and doped *Phaeodactylum tricornutum* cells on conductive indium tin oxide (ITO)-coated glass slides is observed, which can be exploited to generate dielectric biofilms of living microorganisms with luminescent silica shells. In general, this protocol can be envisaged as a profitable route to new functional nanostructured materials for photonics, sensing, or biomedicine via in vivo chemical modification of diatom frustules with organometallic emitters.

**Keywords:** microalgae; diatoms; frustules; biosilica; luminescence; transition metal complexes

## 1. Introduction

Diatom microalgae have recently attracted the interest of materials scientists as they can be envisaged as bio-factories for products of mesoporous biosilica with morphology hardly reproducible by human technologies [1]. Diatom silica cell walls (frustules) are composed of valves and girdles with shape, dimension, and highly periodic nanostructure strictly dependent on the microalgal species [2]. Current literature shows the suitability of diatom frustules or their components for applications in various fields, e.g., as natural photonic crystals and lenses in optics [3,4], biocompatible porous scaffolds for drug delivery and biomedicine [5–9], and nanostructured templates for generation of highly porous conducting materials in electronics [10]. Several approaches for chemical modification of diatom biosilica for specific applications have been developed [11]. Surface chemical modification [12,13] of frustules after cleaning from the organic cell matter represents a common route to biosilica-based materials for sensing or catalysis [14]. The isolation of diatom biosilica shells and their bioclastic conversion [15] into nanostructures with

replicated morphology (but completely different chemical composition) has been used to develop conducting functional nanomaterials for electronics.

Although less widely explored, a further method of functionalization consists in the in vivo incorporation of tailored molecules added to the algal culture. In this case, molecules are up-taken by living cells and conveyed inside their silica deposition vesicles (SDVs) [16], where they are involved in the biosilicification process that generates frustules. This approach has been mainly used to investigate a frustule's morphogenesis, and a rather limited number of fluorescent organic dyes have so far been employed for this aim [17,18].

The benefit of the in vivo incorporation method lies in the possibility to get, in a straightforward way (compared to the genetic engineering approach), biohybrid materials that combine the functions of incorporated molecules with the properties deriving from the frustule nanostructure. A proof of this concept was recently provided by our group through the in vivo incorporation of an aryleneethynylene blue fluorophore into frustules of centric *Thalassiosira weissflogii* diatoms with average 11 μm valve diameter and 20 nm pore size. This yielded luminescent biosilica structures whose photonic properties resulted from the modulation of the dye photoluminescence by the photonic nanopatterned silica structures [19].

We also demonstrated the in vivo incorporation of $[Ir(ppy)_2bpy]^+[PF_6]^-$ complex into frustules of *Thalassiosira weissflogii* diatoms. Hybrid phosphorescent organic/inorganic nanoclusters composed of Ir-doped biosilica nanoparticles trapped within a residual portion of organic cell matter were isolated by ultracentrifugation of doped diatoms after removal of the majority of the organic cell matter with a soft acid/oxidative treatment [20]. In this case, a harsh acid/oxidative ($H_2SO_4$, $HCl/H_2O_2$) protocol, despite being more efficient in purifying the silica shells, was not applied since it degraded the photoluminescent complex. This biotechnological route to luminescent silica nanoparticles is more eco-sustainable than synthetic methods. However, in view of pursuing biohybrid materials for photonics, it would be strongly desirable to achieve a higher purity degree of the metal-doped biosilica, keeping unaltered both the structural properties of the integer silica shells and the photoluminescence of the incorporated organometallic complexes.

We aimed to explore the in vivo incorporation in another microalgal species, *Phaeodactylum tricornutum*, a pennate diatom with raphe dimension of ~12 μm. We also extended the investigation to a series of four organometallic complexes, to evaluate the applicability of our protocol for different luminescent transition metal complexes. *Ph. tricornutum* was selected instead of *Thalassiosira weissflogii* because it is a robust and prolific species providing good amounts of extracted silica per culture batch (more than 100 mg from 5 to 7 mL with a cell density of $10^8$ cells/mL). Its genome, biomineralization, and adhesion properties are well established [21]. Moreover, contrary to the planktonic *Th. weissflogii* diatom, *Ph. tricornutum* is an adherent species forming living biofilms on various substrates, such as conductive indium tin oxide (ITO) glass. This feature deserved our interest because in vivo incorporation of luminescent complexes into *Ph. tricornutum* not only may represent a protocol to produce luminescent biosilica by isolation of diatom frustules after incorporation, but it is also a suitable method to develop luminescent biosilica-based living biofilms adherent to a variety of substrates, with potential applications in photonics. Besides the orange emitting $[Ir(ppy)_2bpy]^+[PF_6]^-$, already investigated for *Thalassiosira weissflogii*, the red phosphor $[Ru(bpy)_3]^{2+} 2[PF_6]^-$ was used as a further model complex with similar octahedral structure, to explore possible effects of a different transition metal ion [Ru(III) vs. Ir(III)] on the incorporation into diatoms. We also extended our study to neutral, commercially available $AlQ_3$ and $ZnQ_2$ complexes, widely used as both electron transporting and green or yellow light emitting materials for optoelectronics, respectively [22–28]. Except for $ZnQ_2$, the complexes were successfully incorporated into frustules, and viability tests on cell cultures revealed a good level of biocompatibility since doped diatoms were still capable of colonizing flat surfaces and forming biofilms. The photoluminescence of incorporated emitters was retained even after harsh acid/oxidative ($H_2SO_4$, $HCl/H_2O_2$) purification

cycles, leading to highly purified valves with integer nanostructure and preserved metal complex photoluminescence.

## 2. Materials and Methods

### 2.1. Materials

Ultrapure grade acetone, ethanol, sulfuric acid, dimethyl sulfoxide, $[Ru(bpy)_3]^{2+}$ $2[PF_6]^-$, $AlQ_3$, and $ZnQ_2$ were purchased from Sigma Aldrich (Germany). $[Ir(ppy)_2bpy]^+$ $[PF_6]^-$ was synthesized according to the literature [29], and its identity was confirmed by $^1H$ NMR spectroscopy. $^1H$ NMR (600 MHz, $CD_3CN$): 8.52 (dt, J = 8.2, 1.1 Hz, 2H), 8.13 (dd, J = 7.9, 1.6, 2H), 8.09–8.03 (m, 2H), 7.98 (ddd, J = 5.5, 1.7, 0.8 Hz, 2H), 7.84 (ddd, J = 8.2, 7.5, 1.5 Hz, 2H), 7.8 (dd, J = 7.9, 1.3 Hz, 2H), 7.6(dd, J = 6.0, 1.7, 0.9, 2H), 7.5 (ddd, J = 7.6, 5.4, 1.2 Hz, 2H), 7.09–6.98 (m, 4H), 6.92 (td, J = 7.4, 1.4 Hz, 2H), 6.28 (dd, J = 7.6, 1.2, 2H) ppm.

### 2.2. Methods

Bidimensional fluorescence images of living diatoms and frustules were recorded by an Axiomat microscope Zeiss (German), using a TRITC-filter set ($\lambda_{exc}$ = 540 nm, $\lambda_{emi}$ > 590 nm) for detection of $[Ru(bpy)_3]^{2+}$ $2[PF_6]^-$ and a combination of FITC-filter ($\lambda_{exc}$ = 465 nm, $\lambda_{emi}$ > 525 nm) and DAPI-filter (blue passband) for $[Ir(ppy)_2bpy]^+[PF_6]^-$, $AlQ_3$, and $ZnQ_2$. A confocal laser scanning microscope, Leica SP8 X, with upright configuration was used for photophysical and morphological characterization; 3D and λ scan images were performed using a UV/diode laser with 405 nm or 488 nm excitation wavelength and a HC PL APO CS2 63x/1.40 oil objective. Emission spectra were recorded in the 450–700 nm spectral range. This interval was divided into 36 detection steps with 5 nm step size. Fourier Transformed Infrared-Attenuated Total Reflectance (FTIR-ATR) spectra were acquired with a Perkin Elmer Spectrum Two Spectrophotometer equipped with A 2 × 2 mm diamond crystal. Spectra were recorded in the range 4000–400 $cm^{-1}$ with a 2 $cm^{-1}$ resolution, using 0.25 $cm^{-1}$ acquisition interval and acquiring 32 scans for each sample. Raman spectra were collected using a LabRAM HR Horiba-Jobin Yvon spectrometer with 532 nm excitation laser under ambient conditions at low laser power (<1 mW) to avoid laser-induced damage. The measuring time was ~200 s. The excitation laser beam was focused through a 50X optical microscope (spot size ~1 mm and work distance 1 cm). The spectral resolution was ~1 $cm^{-1}$.

### 2.3. Diatoms Cultures

*Phaeodactylum tricornutum* diatoms (CCAP strain 1055) were cultured inside a vertical incubator in polystyrene (PS) flasks (250 mL) using sterile seawater mixed with f/2 Guillard solution (buffered with NaOH at pH 7.8) and sodium metasilicate. Growth and adhesion were monitored at 18–22 °C, 65% relative humidity, under a continuous Photon Flux Density (PFD) provided by one cool white fluorescent tube. The light/dark cycle of 14/8 h was employed. Partial medium change was set for every 2 weeks, while total f/2 Guillard solution change was set for every 1 month.

### 2.4. In Vivo Functionalization with Organometallic Emitters

After photostability experiments were performed in triplicate in the diatom medium for in vivo incorporation experiments, a 3 mM stock solution of each organometallic complex in dimethylsulfoxide (1 mL) was prepared. The stock solutions (15 µL) were then diluted with sterile fresh medium (1.5 mL) in 75 mL PS flasks, and after gently stirring, 1.5 mL of the algal culture (cell density 5 × $10^6$ cells/mL) was added to each flask. The in vivo incorporation experiment was carried out using 15 µmol/L concentration of the organometallic complex in the cell culture sample and controlling the culture at 1, 4, and 7 days after the addition of the emitter. For adhesion experiments, 3 mL of algal culture samples was tested using ITO substrates (1 × 1 $cm^{-2}$) in PS multiwells.

### 2.5. Purification of Doped Frustules

After 7 days of doping, glass conic tubes were used to collect and repeatedly wash cells (2200 rpm, 12′). After the last centrifugation, cells were suspended in bidistilled water (500 µL) and treated with a mixture of $H_2SO_4$ (98%, 50 µL), HCl (34%, 50 µL), and $H_2O_2$ (50 µL for 3 times). Cleaning mixtures were kept for 1 h at 55 °C, and the procedure was repeated 3 times. Then, precipitation of doped biosilica occurred by sedimentation. After removal of supernatants, and various washing cycles with distilled water (250–500 µL), all samples were deposited on a glass slide, heated to favor the solvent evaporation, and finally subjected to microscopy observation.

## 3. Results

### 3.1. Photostability of Organometallic Complexes in the Culture Medium

The chemical structures of the luminescent complexes used in this study are reported in Figure 1. Their photostability in the aqueous solution used as the culture medium for the cells was evaluated before starting the staining experiments.

**Figure 1.** Chemical structures of organometallic complexes.

Complexes were dissolved in DMSO (30 µL; 3.37 mM), and the organic solution was poured into f/2 Guillard medium (3 mL) at pH 5.5 [30]. Figure 2a shows the emission spectra recorded immediately after the preparation of all solutions. The resulting emission colors are also shown in the images of solutions in Figure 2b. Figure 2c reports the decay of the intensity of maximum photoluminescence of each complex at 1, 2, 4, and 8 h after the preparation of solutions, with the aim to evaluate the photostability of complexes' emissions in time intervals long enough to let in vivo incorporation start in diatom microalgae.

### 3.2. In Vivo Incorporation of Complexes in Diatoms

*Ph. tricornutum* diatoms were cultured in sterile sodium metasilicate enriched seawater composed of f/2 Guillard solution with nutrients. In vivo staining was carried out according to the procedure reported in the Materials and Method Section. A TRITC filter was used to detect the red emission of both chloroplasts and $[Ru(bpy)_3]^{2+}$, whereas emission of $[Ir(ppy)_2bpy]^+$, $AlQ_3$, and $ZnQ_2$ was observed by a combination of DAPI/FITC filters. Bidimensional fluorescence microscopy images recorded in merge mode for bare and doped *Ph. tricornutum* cells at 1, 4, and 7 days after the addition of complexes are shown in Figure 3.

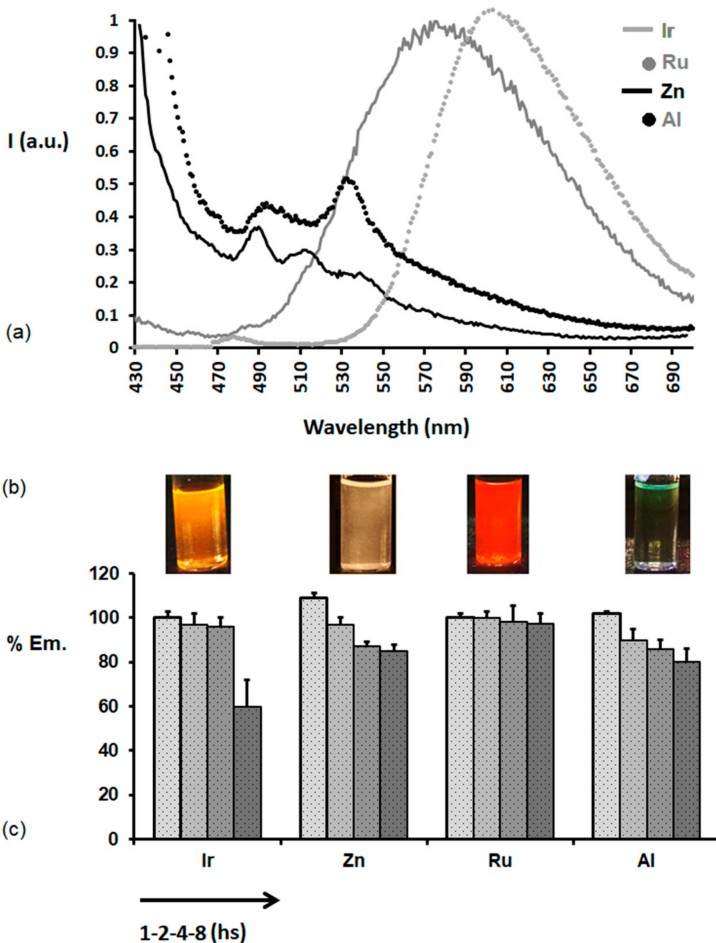

**Figure 2.** (**a**) emission spectra of complexes in f/2 Guillard medium at t = 0; (**b**) images of solutions of complexes in DMSO/f/2 Guillard medium; (**c**) variation of percentages of maximum photoluminescence intensity of complexes' solutions recorded at 0, 1, 2, 4, and 8 h time intervals. Bar error refers to triplicates.

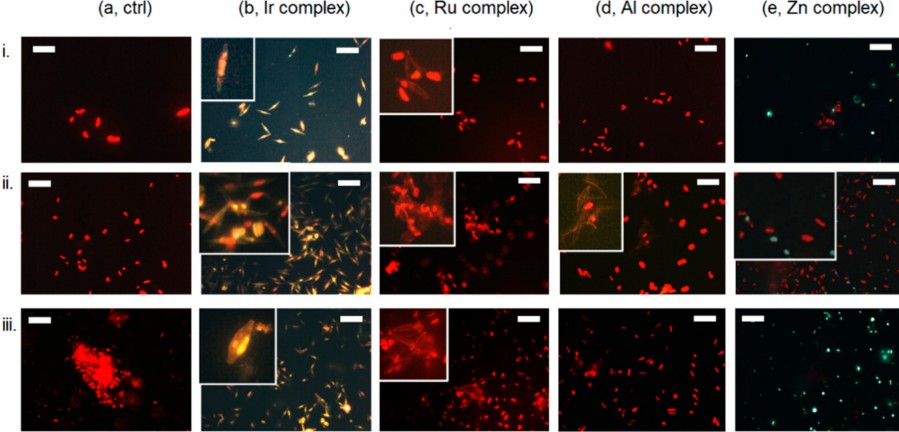

**Figure 3.** Bidimensional fluorescence microscopy images (in merge mode) of *Phaeodactylum tricornutum* (*Ph. tricornutum)* cells in (**a**) the control sample and after addition of (**b**) [Ir(ppy)$_2$bpy]$^+$, (**c**) [Ru(bpy)$_3$]$^{2+}$, (**d**) AlQ$_3$, and (**e**) ZnQ$_2$. Fluorescence micrographs were acquired using red channel ($\lambda_{exc}$ = 540 nm, $\lambda_{emi}$ > 590 nm) for chloroplasts and Ru-complex emission, DAPI-channel ($\lambda_{exc}$= 365 nm, $\lambda_{emi} \geq$ 445 nm) for Ir-emission, and a combination of DAPI-channel and FITC filter ($\lambda_{exc}$ = 465 nm, $\lambda_{emi}$ > 525 nm) for Al- and Zn-emissions. Images are detected after (**i**) 1, (**ii**) 4, and (**iii**) 7 days, and in merge mode when complexes are used. Scale bars: 20 μm. Insets: magnification focus 2×.

### 3.3. Diatoms Adhesion on ITO Substrates

To indirectly assess the viability of *Ph. tricornutum* and at the same time to evaluate their film-forming property, the adhesion of cells on conductive substrates was investigated. Both pristine cells and cells doped with Ir, Ru, and Al complexes adhered to a conductive Indium-Tin Oxide $1 \times 1$ cm$^2$ coated glass slide. Adhesion was analyzed using confocal microscopy, and cells were visualized using bright field (DIC mode). Samples were firstly flushed by pipetting with fresh medium, in order to leave only the strongly adhered cell portion on the substrate surface. Adhesion was monitored at 1, 4, and 7 days (Figure 4).

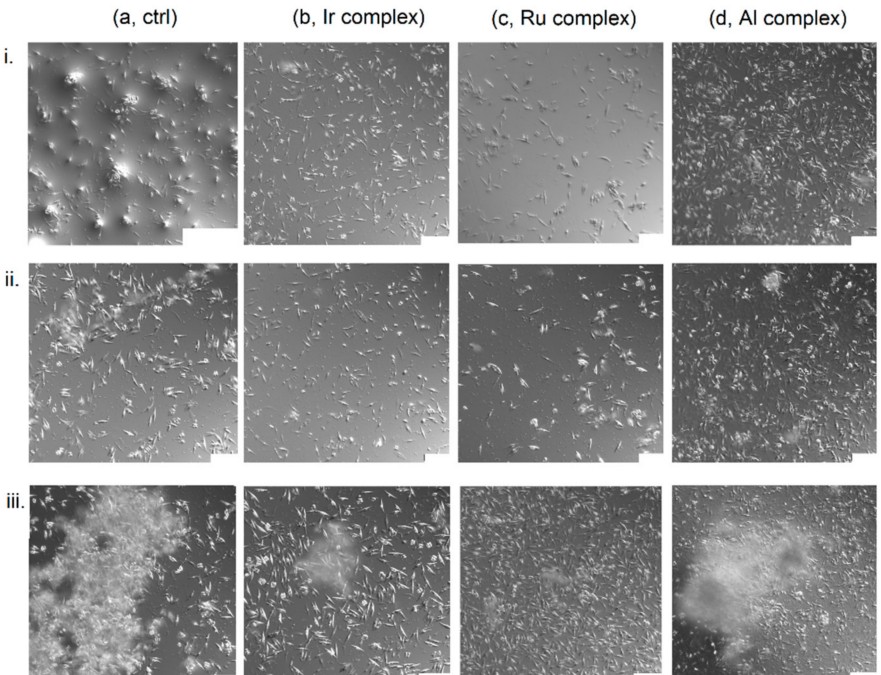

**Figure 4.** Population study via bright field (DIC mode) microscopy for (**a**) bare, (**b**) Ir-doped, (**c**) Ru-doped, and (**d**) Al-doped cells adhered on conductive indium tin oxide (ITO) substrates at (**i**) 1, (**ii**) 4, and (**iii**) 7 days after immersion of the substrate in cell culture samples. Adhesion was valued after washing and gentle flushing of media over substrates. Scale bars: (**ia**) 100 μm; (**ib–d**; **iia–d**; **iiia–d**) 50 μm.

### 3.4. Purification and Characterization of Doped Biosilica

Figure 5i shows confocal microscopy images with 3D reconstruction of living bare and living cells doped with iridum, ruthenim, and aluminum complexes, while Figure 5ii reports fluorescence bidimensional microscopy images of bare and doped shells isolated after harsh acid-oxidative treatment of diatoms, based on the use of sulfuric acid (98%, 50 μL), chloridric acid (34%, 50 μL), and hydrogen peroxide H$_2$O$_2$ (50 μL for 3 times) at 55 °C for 60 min.

Figure 6 shows the photoluminescence spectra recorded, by confocal microscopy, exciting an entire valve of each sample of purified Ir-, Ru-, and Al-doped biosilica and excluding any aggregate. Images of doped valves and neat solid films of complexes were obtained by bidimensional fluorescence microscopy and reported in insets of Figure 6.

The efficacy of the cleaning protocol of biosilica was also demonstrated via Fourier Transformed Infrared-Attenuated Total Reflectance (FTIR-ATR) spectroscopy (Figure 7), and Raman spectroscopy was also suitable to confirm the presence of signals belonging to coordination transitions between metal and non-metal atoms in doped biosilica (Figure 8).

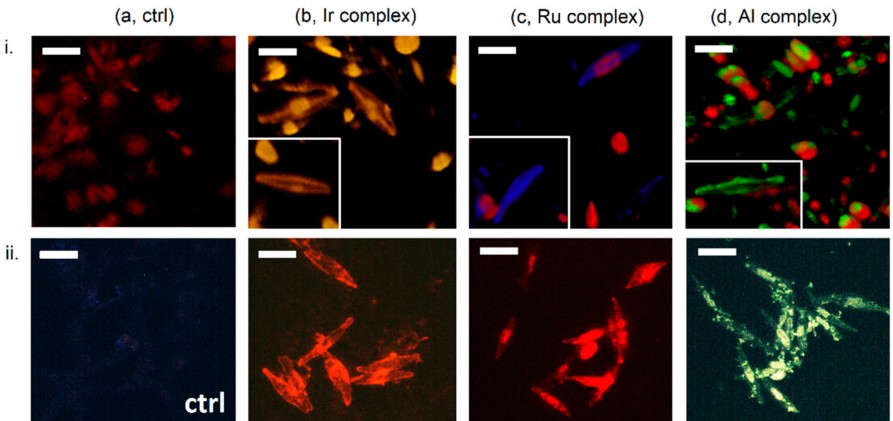

**Figure 5.** (**i**) 3D-confocal reconstruction images of living diatoms (**a**: bare, **b**: Ir-doped, **c**: Ru-doped, **d**: Al-doped cells); (**ii**) bidimensional fluorescence microscopy images of bare and doped frustules after acid-oxidative treatment. Scale bar 10 μm.

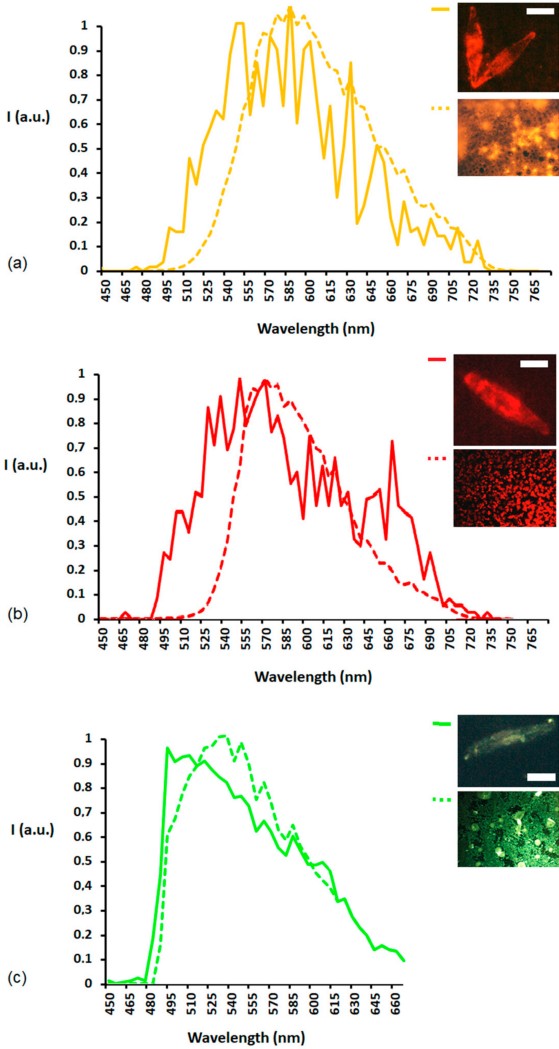

**Figure 6.** Photoluminescence spectra recorded via confocal microscopy of the following: (continuous line) purified (**a**) Ir-doped, (**b**) Ru-doped, and (**c**) Al-doped valves and (dotted lines) neat films of the relative complexes. Inset images report the analyzed valves (top) and the neat films (bottom) of complexes recorded by bidimensional fluorescence microscopy. Scale bar 10 μm.

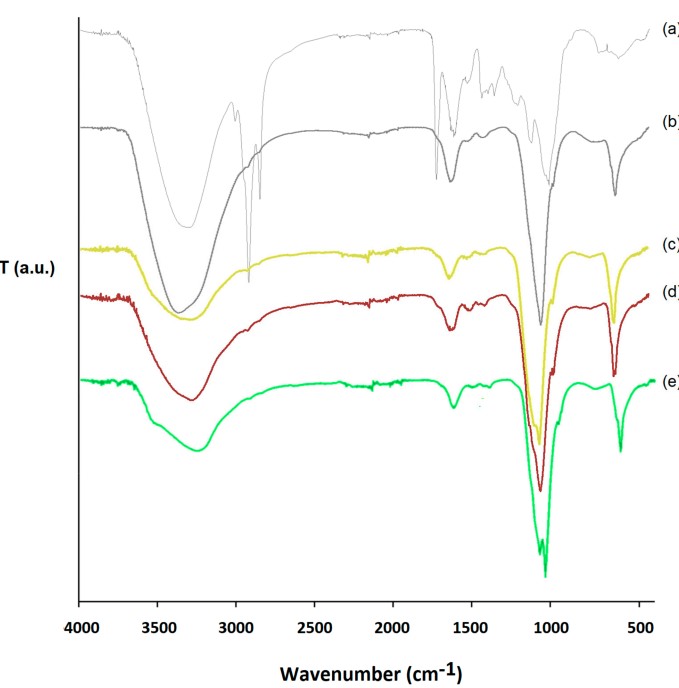

**Figure 7.** ATR spectra of (**a**) dried cells after recovery from culture medium (fresh diatoms) and after purification with acid-oxidative protocol of (**b**) bare, (**c**) Ir-doped, (**d**) Ru-doped, and (**e**) Al-doped shells.

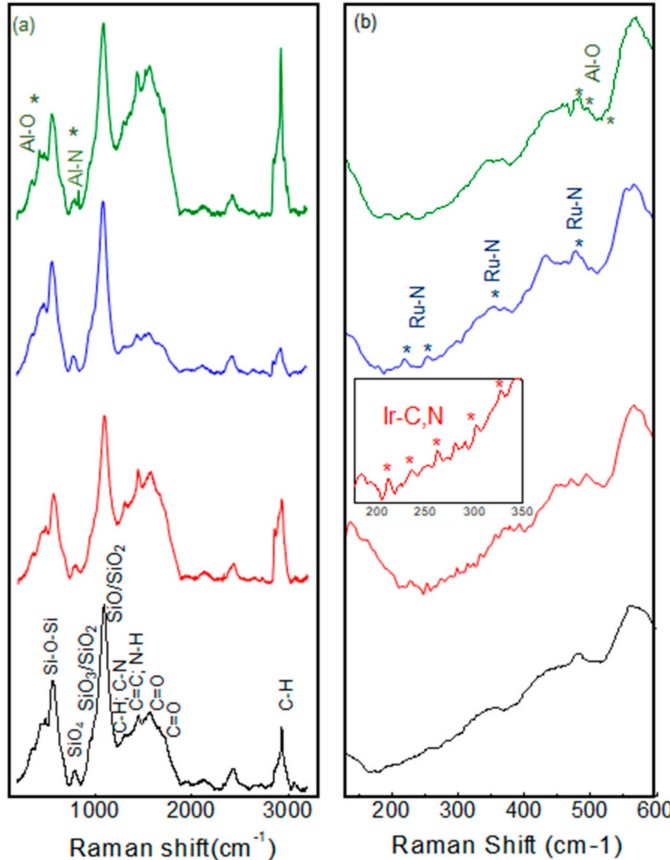

**Figure 8.** Raman spectra of biosilica shells isolated from bare diatoms (grey line), Ir-(red line), Ru-(blue line), and Al-(green line) doped diatoms. (*) indicate metal-C and/or metal-N bands. (**a**) Long range spectra; (**b**) focus on low wavenumber.

## 4. Discussion

The organometallic complexes used as model dopants are reported in Figure 1. Starting from our previous study on centric *Thalassiosira weissflogii* diatoms, we firstly selected $[Ir(ppy)_2bpy]^+[PF_6]^-$ to explore its suitability for doping another microalgal species such as the *Phaeodactylum tricornutum* pennate diatom. Moreover, $[Ru(bpy)_3]^{2+} 2[PF_6]^-$ was selected with the aim of exploring the possibility to extend the incorporation protocol to another complex with analogous cationic octrahedral structure, but with a different nature of transition metal ion. We further investigated the suitability of the doping method for other neutral complexes, such as $AlQ_3$ and $ZnQ_2$, selected as commercially available organometallic dopants [26,27,31] for applications in optoelectronics. Emission spectra recorded for complexes in f/2 Guillard medium show an orange luminescence peaked at 587 nm ($\lambda_{exc}$ = 375 nm) for $[Ir(ppy)_2bpy]^+$, whereas a typical red photoluminescence was observed for $[Ru(bpy)_3]^{2+}$ ($\lambda_{emi}$ = 607 nm, $\lambda_{exc}$ = 463 nm). Yellow and green emissions were recorded for $AlQ_3$ ($\lambda_{emi}$ = 491 nm and 531 nm, $\lambda_{exc}$ = 380 nm) and $ZnQ_2$ ($\lambda_{emi}$ = 489, 512, 535 nm, $\lambda_{exc}$ = 380 nm), respectively (Figure 2a).

The photostability of emissions of the complexes in the cells culture medium was initially demonstrated in the absence of the cells, in an 8 h time period (Figure 2c), long enough to let in vivo incorporation start in diatom microalgae. In fact, complexes mainly started to penetrate inside diatoms in the first day of their addition to cultures. Profiles of emission spectra were retained during the photostability study, with quenching percentages of luminescence intensities after 8 h of $40 \pm 12$ for $[Ir(ppy)_2bpy]^+$, $5 \pm 3$ for $[Ru(bpy)_3]^{2+}$, $19 \pm 6$ for $AlQ_3$, and $15 \pm 3$ for $ZnQ_2$ (Figure 2c).

Considering the limited reduction in photoluminescence intensity and retention of emission spectra profiles, all complexes were explored for feeding experiments. Diatoms were therefore exposed to solutions of organometallic emitters at different concentrations (10, 15 and 20 μM) to evaluate incorporation of the complexes. However, the precipitation of emitters at 20 μM and the reduced photoluminescence at 10 μM limited the study to 15 μM as the suitable value of concentration of all complexes as feeding agents.

Incorporation of the complexes was investigated via bidimensional fluorescence microscopy of bare and doped *Phaeodactylum tricornutum* cells at 1, 4, and 7 days after the addition of the organometallic emitters (Figure 3).

As expected, chloroplasts in untreated cells of the control sample appeared as strongly red emitting mottles that increase in density from day 1 to day 4 of detection, with organization of cells in clusters on day 7 (Figure 3a). Red emitting chloroplasts were also evident in all dispersions of cells treated with organometallic complexes, even 7 days after their addition, thus showing the healthy state of diatoms during the complex incorporation (Figure 3b–e).

Figure 3bi shows that $[Ir(ppy)_2bpy]^+$ started to penetrate in the cells during the first day, and the yellow-orange photoluminescence of the complex was still observed both in protoplasm and in frustules 7 days after its addition (Figure 3bii,biii).

Both $[Ru(bpy)_3]^{2+}$ and $AlQ_3$ were found to penetrate only in frustules (Figure 3c,d and insets). Moreover, in the case of doping with $[Ru(bpy)_3]^{2+}$, an increase in doped cell density was observed passing from 1 to 7 days (Figure 3c). Insets of Figure 3b–d show incorporation of Ir, Ru, and Al emitters into biosilica shells. Conversely, $ZnQ_2$ did not penetrate inside cells, and so we concluded that it is not suitable for in vivo incorporation. This could be in principle due to a strong Zn complex–complex aggregation in solution, leading to a decrease in its bioavailability.

Cell viability was also investigated indirectly by exploring the biological property of *Ph. tricornutum* diatoms to adhere on substrates.

As shown in Figure 4, bare cells increased their density on glass substrates in 4 days, clustering at 7 days of culture. Conversely, iridium and aluminum-doped cells formed smaller clusters due to cell–cell interactions. Ruthenium-doped cells appeared as adherent cells, but they lost the capability of forming cell clusters, likely due to a decrease in cell–cell interaction in the suspension as a possible defense mechanism.

As with all diatom species, *Phaeodactylum tricornutum* cells exhibit a silica cell wall with polysaccharides impregnating the silica matrix. This cell wall protects the inner protoplasm. To confirm the staining of biosilica after the in vivo incorporation of complexes and isolate luminescent frustules, the inner protoplasm, made of chloroplasts and organic matter of cytoplasm, was removed.

Two methods of purification were tested to remove the organic matter and retain the luminescence of complexes into biosilica shells, thus recovering luminescent valves with retained morphology. A soft purification protocol reported by Kroger et al. [32] based on washing with sodium dodecyl sulfate (SDS 5% $V/V$) and ethylene diamino tetraacetic acid (EDTA 100 mM) for 15 min at 60 °C did not successfully remove the chloroplasts and the organic protoplasm from biosilica. Contrary to our previous results reported for *Thalassiosira weissflogii* frustules, a harsh cleaning protocol based on sulfuric acid (98%, 50 µL), chloridric acid (34%, 50 µL), and hydrogen peroxide $H_2O_2$ (50 µL for 3 times) at 55 °C for 30 min turned out to be very efficient to remove the protoplasm and chloroplasts and purify the Ir, Ru, and Al-doped biosilica samples, after in vivo incorporation experiments.

In living diatoms, red chloroplasts only contributed to the red emission of bare cells (Figure 5ai), while the iridium complex stained silica and its emission was also localized in chloroplasts (Figure 5bi). In the case of Ru and Al-doped cells, some empty cells (insets in Figure 5ci,di) and chloroplast-active cells were observed (Figure 5ci,di). After three cycles of acid-oxidative treatment, empty stained valves incorporating the complexes were isolated and spotted over glass slides (Figure 5bii,cii,dii). Control valves exhibited a weak blue autofluorescence mainly due to silica and residual organic matter (Figure 5aii).

Confocal microscopy was also used to record photoluminescence spectra for each sample of doped biosilica after purification, with the aim to compare them with those of neat films of the organometallic complexes. All samples were obtained after 7 days of feeding. Emission spectra were recorded in the 450–750 nm spectral range, exciting ($\lambda_{exc}$ = 405 nm) entire valves. In all cases, the photoluminescence of complexes inside doped valves was slightly blue-shifted versus that of the neat film of the complex (Figure 6a–c). This was likely due to an effect of entrapment of complexes in the silica bulk, or of interaction between the complex photoluminescence and the silica nanostructure, or reduced aggregation compared to the complex as a neat film. As shown in the insets of Figure 7, all purified luminescent valves retained their morphological integrity.

Evidence of the efficacy of the cleaning procedure of biosilica valves was also given by Fourier Transformed Infrared-Attenuated Total Reflectance (FTIR-ATR) spectroscopy (Figure 7). In all samples of purified bare and doped biosilica, as an effect of the acid-oxidative treatment, the –C=O stretching contribution at 1750 cm$^{-1}$ was strongly decreased with respect to living diatoms, whereas the Si-O stretching signal at 1150 cm$^{-1}$ was more evident. The aliphatic –CH symmetric and asymmetric stretching signals, near 3000 cm$^{-1}$, almost disappeared after purification [6].

Raman spectroscopy was also used to check for specific diagnostic signals for coordination transitions between metal and non-metal atoms in doped biosilica (Figure 8). These signals were found in diatom shells treated with organometallic complexes and were absent in untreated biosilica. Figure 8a,b shows Raman spectra of bare biosilica (black line) and Al- (green line), Ru- (blue line), and Ir-doped (red line) shells in two different energy ranges, 100–3300 cm$^{-1}$ and 130–600 cm$^{-1}$, highlight spectral regions where metal-ligand modes could be detected. Two prominent bands were observed in the spectrum of bare biosilica: at ~560 cm$^{-1}$ due to Si-O-Si stretching vibrations, and at ~1100 cm$^{-1}$ due to $SiO/SiO_2$ symmetric stretching. Shoulders at 350, 480, and 950 cm$^{-1}$ were attributed to O-Si-O deformation, $O_3SiOH$ tetrahedral vibrations, and $SiO_3/SiO_2$ symmetric stretching, respectively [33]. The bands at ~800 cm$^{-1}$ were due to $SiO_4$ symmetric stretching. In the 1250–1900 cm$^{-1}$ range, $-CH_2$, $-CH_3$, and -C-C- bands were evident while the C-H stretching vibration appeared in the range between 2800 and 3000 cm$^{-1}$ due to an incomplete removal of aromatic and aliphatic carbon sources associated to biosilica even after the cleaning

treatment. The Raman spectrum of shells doped with Al complex exhibited several modes in the range of 150–1000 cm$^{-1}$, which were assigned to Al-O vibrations, at ~480, ~500, and 520 cm$^{-1}$ and to Al-N at 830 cm$^{-1}$ [34,35]. Four metal-ligand modes could be observed for Ru-doped biosilica at 225, 255, 340, and 470 cm$^{-1}$ [36,37]. For Ir-treated shells, ν (Ir-C, N) bands were very weak and could be detected in the range 193–324 cm$^{-1}$. The inset in Figure 8 highlights ν (Ir-C, N) bands [38].

## 5. Conclusions

In conclusion, we have investigated in vivo incorporation of iridium, ruthenium, aluminum, and zinc luminescent organometallic complexes into *Phaeodactylum tricornutum* diatom microalgae. Except for ZnQ$_2$, the complexes penetrate cells and stain frustules without altering diatom viability even 7 days after their addition. The microalgal species still retains its capability of adhesion by cell–cell and cell–substrate interactions after the uptake of the organometallic emitters, thus forming luminescent biofilms of living microorganisms. Moreover, a hard acid-oxidative treatment—based on sulfuric acid, chloridric acid, and hydrogen peroxide—of doped diatoms allows isolation of biosilica valves while retaining both their pristine nanostructured morphology and the emission properties of the incorporated transition metal complex. The evidence of incorporation of complexes into frustules was provided by confocal and bidimensional fluorescence microscopies as well as by FTIR-ATR and Raman spectroscopies. This biotechnological protocol represents a straightforward method of preparation of biosilica doped with luminescent organometallic complexes, with the benefit of easily achieving silica nanostructures, never reported so far, that combine photoluminescence of organometallic complexes with structural properties of diatom valves optimized by nature over a billion years. Moreover, the film-forming properties of the semi-benthonic diatom can be exploited to self-assemble in vivo thin luminescent biosilica-based biofilms that may have potential application in photonics. Although the reasons why ZnQ$_2$ is not suitable as dopant for *Phaeodactylum tricornutum* are not clarified, our results show the suitability of the method for preparation of light-emitting iridium, ruthenium, and aluminum-doped biosilica valves with integer nanostructure, and can pave the way to a variety of biohybrid materials for photonics and optoelectronics.

**Author Contributions:** D.V. contributed to diatom culture and feeding experiments; R.R. contributed to complex synthesis and characterization, experimental design, and paper writing; E.A. and P.A. performed confocal microscopy analysis; M.M.G. contributed to Raman analysis; S.R.C. contributed to the spectroscopic characterization of biosilica; G.M.F. contributed to planning all the experiments and paper writing. All authors have read and agreed to the published version of the manuscript.

**Funding:** This research was funded by the European Commission through EU project 800926-HyPhOE (Hybrid Electronics based on Photosynthetic Organisms), H2020-MSCA-ITN-2019 project 860125—BEEP (Bio-inspired and bionic materials for enhanced photosynthesis).

**Institutional Review Board Statement:** Not applicable.

**Informed Consent Statement:** Not applicable.

**Data Availability Statement:** Data are contained within the article.

**Conflicts of Interest:** The authors declare no conflict of interest.

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
