# Peer review of "Light-Emitting Biosilica by In Vivo Functionalization of Phaeodactylum tricornutum Diatom Microalgae with Organometallic Complexes"

_applsci, doi:10.3390/app11083327_

Round 1

Reviewer 1 Report

In the manuscript entitled "Light emitting biosilica by in vivo functionalization of Phaeodactylum tricornutum diatom microalgae with organometallic complexes", the authors present their work incorporating various photoluminescent organometallic complexes into diatom frustules as a pathway for the creation of novel nanostructured silica-based luminescence emitters. Overall, the authors presented their main points quite well. Clarifying a few minor points could help improve the message:

  • The decision to use Phaeo is not clear. The transition between previous work on T. weissflogii and the new “more general protocol” work is difficult to follow, especially for readers not familiar with diatoms.
  • On the same theme, the distinction between live diatoms vs. isolated biosilica frustules also should be stated in consistent terms. Non-experts in the field will have a hard time grasping the distinction in places.
  • Figure legends could be more descriptive.
    • Sometimes the wording of the legends makes the figures more difficult to interpret.
    • When describing the fluorescence filters for the microscopy images, the authors may wish to consistently describe them by the bandpass values. Not all commercially available filters are alike, and reiterating the wavelengths is the more accurate way to describe them.
    • In Figure 3, scale bars should be included in more than one image (especially when the figure legend states that “scale bars” - plural - are present). Also, how do the inset windows vary in terms of magnification?
    • In Figure 4, please note which scale bars are which length.
    • In Figure 6, scale bar lengths are not noted in the legend.

Author Response

Point-by-point response to Reviewer 1

Revisions made according to suggestions of reviewer 1 have been yellow highlighted in the main text.

Reviewer 1:

In the manuscript entitled "Light emitting biosilica by in vivo functionalization of Phaeodactylum tricornutum diatom microalgae with organometallic complexes", the authors present their work incorporating various photoluminescent organometallic complexes into diatom frustules as a pathway for the creation of novel nanostructured silica-based luminescence emitters. Overall, the authors presented their main points quite well. Clarifying a few minor points could help improve the message:

  1. The decision to use Phaeo is not clear. The transition between previous work on T. weissflogii and the new “more general protocol” work is difficult to follow, especially for readers not familiar with diatoms. On the same theme, the distinction between live diatoms vs. isolated biosilica frustules also should be stated in consistent terms. Non-experts in the field will have a hard time grasping the distinction in places.

Revision 1: According to the reviewer’s comment, we have revised the text at page 2 (lines 74-87), page 12 (lines 302-302, 309-313), page 14 (lines 387-389) to explain the reasons why we have used Phaeodactilum instead of Thalassiosira and to clarify distinction between living diatoms and isolated luminescent frustules.

Page 2 (lines 74-87): “Here we aim to explore the in vivo incorporation in another microalgal species, Phaeodactylum tricornutum, a pennate diatom with rafe dimension of ~12 μm. We have also extended the investigation to a series of four organometallic complexes, to evaluate the applicability of our protocol for different luminescent transition metal complexes. Ph. tricornutum was selected instead of Thalassiosira weissflogii because it is a robust and prolific specie providing good amounts of extracted silica per culture batch (more than 100 mg from 5-7 mL with a cell density of 108 cells/mL). Its genome, biomineralization and adhesion properties are well established [21]. Moreover, contrary to the planktonic Th. weissflogii diatom, Ph. tricornutum is an adherent specie forming living biofilms on various substrates, such as conductive ITO glass. This feature deserved our interest since in vivo incorporation of luminescent complexes into Ph. tricornutum not only may represent a protocol to produce luminescent biosilica by isolation of diatoms’ frustules after incorporation, but it is also a suitable method to develop luminescent biosilica-based living biofilms adherent to a variety of substrates, with potential applications in photonics.”

Page 12 (lines 302-302): “Cells viability was also investigated indirectly exploring the biological property of Ph. tricornutum diatoms to adhere on substrates.”

Page 12 (lines 309-313): “As all diatom species, Phaeodactylum tricornutum cells exhibit a silica cell wall with polysaccharides impregnating the silica matrix. This cell wall protects the inner protoplasm. To confirm the staining of biosilica after the in vivo incorporation of complexes and isolate luminescent frustules, the inner protoplasm, made of chloroplasts and organic matter of cytoplasm, was removed.”

page 14 (lines 387-389): “Moreover, the film forming properties of the semi-benthonic diatom used can be exploited to self-assemble in vivo thin luminescent biosilica based biofilms that may have potential application in photonics.”

  1. Figure legends could be more descriptive. Sometimes the wording of the legends makes the figures more difficult to interpret. When describing the fluorescence filters for the microscopy images, the authors may wish to consistently describe them by the bandpass values. Not all commercially available filters are alike, and reiterating the wavelengths is the more accurate way to describe them.

Revision 2: Captions of Figures 3, 4, 5, 6, 8 have been revised, including more details, information on filters and scale bars. In the caption of Figure 2 we have detailed that Bar error refers to triplicates.

  1. In Figure 3, scale bars should be included in more than one image (especially when the figure legend states that “scale bars” - plural - are present). Also, how do the inset windows vary in terms of magnification? In Figure 4, please note which scale bars are which length. In Figure 6, scale bar lengths are not noted in the legend.

Revision 3: We have added scale bars and details of inset magnification in Figures 3, 4 and 6 as suggested.

Reviewer 2 Report

The authors provide a new approach for in vivo incorporation of a series of organometallic photoluminescent complexes in Phaeodactylum tricornutum diatom cells. The resulting luminescent silica nanostructures have a high potential for use in photonics, sensing or biomedicine. The idea to clean the frustules by a more harsh method than in former work is very helpful in producing new functional nanostructured materials.

All in all, the work is new, has a good quality and is worth to publish in “Applied Sciences”, but a few things have to be clarified or amended in the manuscript. These are the following:

  • Generally, the authors cite a lot of scientific work, but an appreciable part of this is their own work. However, there are more papers in literature worth to cite.
  • It is not clear, why the authors now use Phaeodactylum tricornutum instead of Thalassiosira weissflogii preferred in their former work. The authors should clarify this and give the reader an idea about differences between these two species.
  • The authors state that the complex [Ir(ppy)2bpy]+ [PF6]- was synthesized according to the literature. They should demonstrate that the synthesis led to the desired result.
  • In the methods section more details about raman measurement, e.g. laser power, measuring time, spectral resolution … are needed.
  • In the section „Purification of doped frustules“ a few values have to be clarified: H2SO4 (50 μL – which concentration??), HCl (34%, 50 μL) and H2O2 (3x50 μL – what does it mean:3x50 µl?).
  • Figure 1: The use of dotted and solid lines in the complex structures seems to be chemically not correct.
  • Figure 2: Where are the error bars from? Please describe the experiment used.
  • Figure 3: To prove the statement in the discussion section, that the complex is completely included by the diatoms, the images of the start of the experiment (time 0) are necessary.
  • Lines 205 and 206: These values don’t fit the values mentioned in the experimental section! See also line 300!
  • In Figure 5 the dimension of scale bar has to be given.
  • The photoluminescence spectra of Ir- and Ru-doped frustules in Figure 6 do not fit to the emission spectra of complexes in Figure 2, the wavelengths of maxima are shifted and for Ir and Ru even transposed. Is it true? What is the reason for this?
  • Figure 7: In IR spectra you don’t have the intensity on ordinate but the transmission.
  • Line 252: The authors state that “complexes mainly penetrate inside diatoms in the first day of their addition to cultures.” How can you support this thesis?
  • Evaluation of IR spectra: There are some changes due to complexes, mainly in the Si-O stretching region, e.g. a shift and a split of this band. It would be really interesting to see this region enlarged and to interpret these changes.
  • Raman spectra: The assignment of very week complex bands is here based on values in literature. It would be more convenient to measure the raman spectra of pure complexes and compare them with the spectra of produced materials.
  • In general, the authors mainly use the word incorporation. In my opinion, there is no proof in this paper for a really incorporation of the complexes into the biosilica. This has to be proofen by former analytical methods. In this case the complexes may only be attached to the surface by adsorption processes or weak physical bonding or associated with the biosilica. This has to be clarified!
  • What do you know about the mechanism of uptake of complexes? Does it really take place only in the starting of cultivation? The growth of diatoms goes on further, would it make sense to offer more and more complex?

Author Response

See the attachment below for the complete response.

Point-by-point response to Reviewer 2

Revisions made according to suggestions of Reviewer 2 have been yellow highlighted in the main text.

Reviewer 2:

The authors provide a new approach for in vivo incorporation of a series of organometallic photoluminescent complexes in Phaeodactylum tricornutum diatom cells. The resulting luminescent silica nanostructures have a high potential for use in photonics, sensing or biomedicine. The idea to clean the frustules by a more harsh method than in former work is very helpful in producing new functional nanostructured materials. All in all, the work is new, has a good quality and is worth to publish in “Applied Sciences”, but a few things have to be clarified or amended in the manuscript. These are the following:

  1. Generally, the authors cite a lot of scientific work, but an appreciable part of this is their own work. However, there are more papers in literature worth to cite.

Revision 1: We have revised references according to the Reviewer’s suggestion. Hence, we have substituted ref. 1, 5, 7, 12, 13, 15, 23, 25.

  1. It is not clear, why the authors now use Phaeodactylum tricornutum instead of Thalassiosira weissflogii preferred in their former work. The authors should clarify this and give the reader an idea about differences between these two species.

Revision 2:  According to the reviewer’s comment, we have revised the text adding some sentences at pag.2, to explain the reason why we have used Phaeodactilum instead of Thalassiosira in this paper. The added sentences are (lines 74-87):

Here we aim to explore the in vivo incorporation in another microalgal species, Phaeodactylum tricornutum, a pennate diatom with rafe dimension of ~12 μm. We have also extended the investigation to a series of four organometallic complexes, to evaluate the applicability of our protocol for different luminescent transition metal complexes. Ph. tricornutum was selected instead of Thalassiosira weissflogii because it is a robust and prolific specie providing good amounts of extracted silica per culture batch (more than 100 mg from 5-7 mL with a cell density of 108 cells/mL). Its genome, biomineralization and adhesion properties are well established [21]. Moreover, contrary to the planktonic Th. weissflogii diatom, Ph. tricornutum is an adherent specie forming living biofilms on various substrates, such as conductive ITO glass. This feature deserved our interest since in vivo incorporation of luminescent complexes into Ph. tricornutum not only may represent a protocol to produce luminescent biosilica by isolation of diatoms’ frustules after incorporation, but it is also a suitable method to develop luminescent biosilica-based living biofilms adherent to a variety of substrates, with potential applications in photonics.

  1. The authors state that the complex [Ir(ppy)2bpy]+ [PF6]- was synthesized according to the literature. They should demonstrate that the synthesis led to the desired result.

Revision 3: At page 3 (lines 107-111) we have added the digitalized signals of the 1H NMR spectrum recorded for the synthesized complex, that are in accordance with those reported for this emitter in the literature (ref.29).

  1. In the section „Purification of doped frustules“ a few values have to be clarified: H2SO4 (50 μL – which concentration??), HCl (34%, 50 μL) and H2O2 (3x50 μL – what does it mean:3x50 µl?). Lines 205 and 206: These values don’t fit the values mentioned in the experimental section! See also line 300!

Revision 4: We have corrected the sentence at page 3 (lines 157-159) as it follows: “After the last centrifugation, cells were suspended in bidistilled water (500 μL) and treated with a mixture of H2SO4 (98%, 50 µL), HCl (34%, 50 µL) and H2O2 (50 µL for 3 times). We have also revised the sentences at page 7 (lines 224-225) and page 12 (lines 322-323).

  1. Figure 1: The use of dotted and solid lines in the complex structures seems to be chemically not correct.

Revision 5: We have modified the chemical structures according to the Reviewer’s comment

  1. Figure 2: Where are the error bars from? Please describe the experiment used.

Revision 6: We have specified the use of triplicate batches in the caption of Figure 2 and in the experimental.

  1. Figure 3: To prove the statement in the discussion section, that the complex is completely included by the diatoms, the images of the start of the experiment (time 0) are necessary.

Revision 7: Our aim is not to demonstrate that the complex is completely transferred from solutions to diatoms but only that the dye accumulation starts within the first 24 hours. To answer to the Reviewer comment, here we report the images that refer to the starting time of the experiment, taken 2h after the incubation of diatoms with organometallic complexes and subsequent washing steps of the dyes-enriched supernatants. As shown, only Ir complex barely stains Phaeo shells after 2 hours. We don’t consider appropriate to include these pictures in Figure 5 and we prefer to discuss results starting from 24 hours.

  1. In Figure 5 the dimension of scale bar has to be given.

Revision 8: The scale bar dimension has been added in the caption of Figure 5.

  1. The photoluminescence spectra of Ir- and Ru-doped frustules in Figure 6 do not fit to the emission spectra of complexes in Figure 2, the wavelengths of maxima are shifted and for Ir and Ru even transposed. Is it true? What is the reason for this?

Revision 9: Figure 2 reports luminescence spectra of the organometallic complexes in solution of f/2 Guillard medium, while Figure 6 shows spectra referred to neat solid films of organometallic complexes (dotted lines) and spectra of dried biosilica stained with complexes. The differences between photoluminescence spectra in solution and in neat solid films are not unusual: the emission profiles change depending on solvents used for solution while photoluminescence of solid films can be dependent on formation of aggregated states of the emitting complex.

  1. Figure 7: In IR spectra you don’t have the intensity on ordinate but the transmission.

Revision 10: Figure 7 has been revised accordingly.

  1. Line 252: The authors state that “complexes mainly penetrate inside diatoms in the first day of their addition to cultures.” How can you support this thesis?

Revision 11: We have modified the sentence at page 12 (lines 271-272) as: “complexes mainly start to penetrate inside diatoms in the first day of their addition to cultures.” We support this thesis considering the feeding experiments performed at 2 hours (see revision 7) and 24 hours after the addition of complexes.

  1. Evaluation of IR spectra: There are some changes due to complexes, mainly in the Si-O stretching region, e.g. a shift and a split of this band. It would be really interesting to see this region enlarged and to interpret these changes.

Revision 12: The IR signals can be dependent on many factors when biosilica is produced and extracted from living diatoms. Variations in time, temperature and concentrations of the cleaning procedures, together with small variations in culture conditions (which physiologically occur) can interfere with IR signals, including the Si-O stretching signal. Hence, we cannot consider the Si-O shifts strictly related to the presence of small amounts of complexes, and we use IR-spectroscopy as a tool to control the efficacy of the cleaning method of biosilica.

  1. Raman spectra: The assignment of very week complex bands is here based on values in literature. It would be more convenient to measure the raman spectra of pure complexes and compare them with the spectra of produced materials. In the methods section more details about raman measurement, e.g. laser power, measuring time, spectral resolution … are needed.

Revision 13:  To complete the experimental details, we have added the following sentence at pages 3-4 (lines 128-131): “under ambient conditions at low laser power (<1mW) to avoid laser-induced damage. The measuring time was ~200sec. The excitation laser beam was focused through a 50X optical microscope (spot size ~1 mm and work distance 1 cm). The spectral resolution was ~1 cm-1.”

Figure 8 in the paper has been replaced by a new one where metal-C and/or metal/N bands are highlighted by (*) symbols and the Raman spectrum of diatoms shells treated with Al-complexes has been replaced with a new one where Al-N and Al-O bands are more evident. We use (*) symbols in the figure to indicate the Raman bands that we experimentally found and showed in Figure 8 in the paper for diatoms shells treated with organometallic complexes.

In lines 365-370 comments to Figure 8 have been slightly modified and the caption of Figure 8 has been completed with information of (*) indicating metal-C and/or metal-N bands.

We have recorded the Raman spectra of pure complexes, and the results are reported below. They are consistent with those of similar signals reported in the literature. However, in our opinion, their insertion and description in the manuscript may be heavy reading and, for this reason, we would prefer to limit discussion comparing the spectra of luminescent biosilica to reference signals from the literature.

To record Raman spectra of pure complexes, we have drop casted Ir, Ru and Al complexes on glass substrates using DMSO as the solvent. The following figures show the Raman spectra of pure complexes in different range to highlight the metal-ligand bands.

Specifically, Raman spectra for all the complexes show bands in the range 150-213 cm-1 due to the phenyl or pyridyl ring in-plane twisting modes and due to the metal-ligand  (M-L) bending, denoted as D(ring-ring). In the ranges 279-290 cm-1 and 419-448 cm-1 there are bands due to the phenyl or pyridyl ring out-of-plane twisting modes, denoted as G(ring-ring) or G(ring), whereas the ring-ring stretching, n(ring-ring) is observed in the range 374-383 cm-1. In the Raman spectrum of Al complexes, there are also bands in the range 506-570 cm-1 due to the  G(ring) mode and in the range 631-673 cm-1 due to the D(ring) mode.

Concerning the Raman spectrum of pure Al-complex, we found the Al-N modes at ~250 cm-1, ~305 cm-1 and ~430 cm-1 and ~822 cm-1 and Al-O modes at ~325 cm-1, ~410 cm-1, ~480 cm-1, ~503 cm-1, ~520 cm-1, ~750 cm-1 and ~822 cm-1. The Al-O and Al-N vibrations come from different references [Lv Jinlong et al. Applied Surface Science 273 (2013) 192-198; I.C. Oliveira et al. Journal of Materials Science Materials in Electronics 12 (2001) 259-262;  A. V. Sergeeva, et al. Minerals 10 (2020) 781 1-14; L.E. McNeil et al. Journal of the American Ceramic Society 76 (1993) 32- 36; S. Martínez-Ramírez et al. Cement and Concrete Composites 73 (2016) 251-256; H. D. Ruan, J. Raman Spectrosc. 2001; 32: 745–750].

Concerning the Raman spectrum of pure Ru-complex, we found five Ru-N modes at ~235 cm-1, at ~260 cm-1, at ~335 cm-1, at ~375 cm-1 and at ~450 cm-1. The Ru-N vibrations are consistent with data from literature [S.-H. Lai et al. J. Raman Spectrosc. 2011, 42, 332–338; Stephen K. Doom et al. J. Am. Chem. Soc. 1989, 11 1, 4704-47 12].

Finally, concerning the Raman spectrum of pure Ir-complex, we found seven Ir-N,C modes at ~195 cm-1, at ~235 cm-1, ~257 cm-1, ~270 cm-1, ~310 cm-1, ~320 cm-1  and ~332 cm-1. The Ru-N vibrations are consistent with data from literature [S.-H. Lai, et al. J. Raman Spectrosc. 2011, 42, 332–338].

  1. In general, the authors mainly use the word incorporation. In my opinion, there is no proof in this paper for a really incorporation of the complexes into the biosilica. This has to be proofen by former analytical methods. In this case the complexes may only be attached to the surface by adsorption processes or weak physical bonding or associated with the biosilica. This has to be clarified!

Revision 14: In general, the word incorporation is used to indicate the uptake of a functional molecule (in this case a visible light emitting complex) by the living diatom and its inclusion in vivo in the silica shell. This word is used mainly to distinguish that the complex has not been grafted directly by a chemical reaction to the surface of isolated biosilica shells. In general, it is not possible to assess the nature of chemical binding of complexes to the silica after the in vivo uptake, but the observation that the complex photoluminescence is still observed after cells reproduction suggests that, although we don’t know the nature of interaction with silica, the emitting complex may take part to the valve morphogenesis of a new cell, this meaning that it somehow interacts with silica network and not only on its surface.

  1. What do you know about the mechanism of uptake of complexes? Does it really take place only in the starting of cultivation? The growth of diatoms goes on further, would it make sense to offer more and more complex?

We have not explored the mechanism of uptake of these complexes: in our opinion, the positively charged complexes may be uptaken by electrostatic interaction with silica, but we haven’t experimental evidence of this hypotesis. However, the persistence of cells staining over time suggests at least an aspecific mechanism of accumulation that increases the concentration of the dye in the cells.

The uptake of complexes does not occur only in the starting period of the complex addition to the culture. We have revised the manuscript assessing that the uptake mainly starts in the first 24 hours and the complexes in solution are not completely uptaken by the culture (lines 271 and 291-293). For this reason, in our experiments it has not been necessary to add more and more complex.

Round 2

Reviewer 2 Report

The authors addressed all my suggestions in review and in my opinion the paper is now ready for publication.